# Indirect Calorimetry in Clinical Practice

**DOI:** 10.3390/jcm8091387

**Published:** 2019-09-05

**Authors:** Marta Delsoglio, Najate Achamrah, Mette M. Berger, Claude Pichard

**Affiliations:** 1Clinical Nutrition, Geneva University Hospital (HUG), 1205 Geneva, Switzerland; 2Nutrition Department, Rouen University Hospital Center, 76000 Rouen, France; 3Service of Intensive Care Medicine & Burns, University of Lausanne Hospitals (CHUV), 1005 Lausanne, Switzerland

**Keywords:** indirect calorimetry, indirect calorimeter, resting energy expenditure, nutrition therapy

## Abstract

Indirect calorimetry (IC) is considered as the gold standard to determine energy expenditure, by measuring pulmonary gas exchanges. It is a non-invasive technique that allows clinicians to personalize the prescription of nutrition support to the metabolic needs and promote a better clinical outcome. Recent technical developments allow accurate and easy IC measurements in spontaneously breathing patients as well as in those on mechanical ventilation. The implementation of IC in clinical routine should be promoted in order to optimize the cost–benefit balance of nutrition therapy. This review aims at summarizing the latest innovations of IC as well as the clinical indications, benefits, and limitations.

## 1. Introduction

The accurate determination of patients’ energy needs is required to optimize nutritional support and to reduce the deleterious effects of under- and over-feeding [1]. However nutritional requirements may be difficult to predict, especially in patients with acute or chronic conditions where many factors (e.g., stress, brain activity, endocrine profile, inflammation status, feeding state, drugs, etc.) affect their resting energy expenditure (REE). REE represents the energy expended by the body during a 24 h non-active period to maintain involuntary functions such as substrate turnover, respiration, cardiac output, and body temperature regulation [2]. In healthy sedentary adult subjects REE constitutes two-thirds of the total energy expenditure (TEE) which includes energy needed for nonobligatory expenses such as physical activity and diet-induced thermogenesis. This assumption is probably not entirely correct in sick patients where continuous nutrition, treatments and diseased-related stress increase, or decrease REE: extreme variations among individuals can be observed within the same day [3]. 

Indirect calorimetry (IC) is considered as the gold standard to measure REE, by measuring oxygen consumption (VO_2_) and carbon dioxide production (VCO_2_). Apart from REE, other parameters can be derived from IC, such as substrate (carbohydrates, fat, and protein) utilization. Indeed, the ratio between VCO_2_ and VO_2_ ($\frac{{\mathrm{VCO}}_{2}}{{\mathrm{VO}}_{2}}$) defines the respiratory quotient (RQ) that corresponds to the substrate use. The complete oxidation of glucose generates a RQ value of 1.0, while a RQ of 0.7 is indicative of a mixed substrate oxidation [2]. 

Recent technical developments of indirect calorimeters allow accurate, non-invasive, and easy IC measurements in spontaneously breathing patients as well as in those on mechanical ventilation. Recent trials have shown IC allows clinicians to personalize the prescription of nutrition support to the metabolic needs and to monitor the metabolic response to nutrition therapy promoting a better clinical outcome in acutely ill patients [4,5,6,7,8]. These results have increased the interest for IC as a tool for improving the routine nutritional evaluation and prescription.

This review aims to summarize the latest innovations of IC as well as the clinical indications, benefits, and limitations of the technique.

## 2. Indications

Equations predicting REE are relatively reliable in healthy subjects. Contrariwise, in case of disease or trauma, REE is influenced by many factors with synergic or antagonist impact (Table 1) [1,2,9]. In these conditions, predictive equations are largely inaccurate [10]. As a consequence, IC is mainly indicated in three different scenarios: (a) clinical conditions significantly modifying REE; (b) failure of a nutrition support based on predicted energy needs to maintain or restore body weight; (c) acute critical illness associated with large and dynamic changes of metabolic stress level [11]. The latter indication requires special caution, as the IC results will reflect the patient’s instability: repeated IC studies are required to observe the evolution of the metabolic response to hemodynamic instability, fever, surgery, weaning from mechanical ventilation, etc. A single IC is rarely sufficient, thereby repeated measurements every second or third day are needed in patients with rapid changes of their clinical status [12,13,14]. 

Performing IC in clinical routine is recommended not only to set an optimal nutrition therapy, but also to monitor the results of the nutritional interventions, in order to avoid complications of inappropriate nutrition, i.e., of under- or over-feeding. 

## 3. Practicalities

IC allows measuring REE in both mechanically ventilated and spontaneously breathing patients (Figure 1). In mechanically ventilated patients, the respiratory gas sampling is acquired by the circuit connecting the endotracheal tube to the ventilator and measured by ‘breath-by-breath’ or mixing chamber analyses. In spontaneously breathing subjects, a ventilated canopy hood or a fitted face-mask is used to collect inspired and expired gas. In canopy system the patient’s head is surrounded by a clear rigid hood and a pump pulls air through the canopy at a constant rate. In both ventilator and canopy mode, respiratory gases are analyzed by the indirect calorimeter and used to calculate REE through Weir’s equation [1]. As will be detailed below, acutely and chronically ill patients at high risk of malnutrition are the ones who benefit most from IC. 

### 3.1. Acute Diseases

Metabolic responses to shock and injury were first described by Sir Cuthbertson [15]. According to his investigations, there is a short ebb phase starting immediately after a traumatic shock and followed by a flow phase of longer duration after 3 to 10 days (Figure 2) [16]. The ebb phase is characterized by a decrease in metabolic rate, oxygen consumption, body temperature, and enzymatic activity. The flow phase, on the contrary, is marked by an increased catabolism, with a high oxygen consumption and an elevated REE rate. 

Recent studies have supported this pattern of longitudinal changes in REE in acutely ill patients [17,18,19]. The degree of increase from normal REE may reflect the severity of the metabolic response to the injury. However, the interaction between natural course of disease, individual inflammatory and immune system response, and medical treatments make it difficult for the clinician to evaluate caloric needs. For these reasons, IC is the only available tool enabling the individual assessment of the patients’ energy needs, and to ensure precise nutritional interventions in acute conditions such as post-surgery period, pancreatitis, kidney injury, and sepsis.

The post-operative period of uncomplicated surgery has been associated with a modest 7% increase in energy metabolism as the effect of the surgical operation itself, that cannot be predicted by static equations [20]. Non-septic patients with acute pancreatitis presented a hypermetabolic state with a REE increase of 120 ± 11% compared to predicted [21]. Energy metabolism in patients with renal failure has been studied as well. Acute kidney injury (AKI) has not been shown to affect REE, but concomitant conditions, such as sepsis, mostly play role in the hypermetabolism found in these patients [22,23]. The presence of hypermetabolism has also been associated with lower age and higher vasoactive drug dose [24] and no predictive formula was able to correctly calculate energy needs in AKI patients compared to IC [25]. Sepsis is characterized by a hyperdynamic cardiovascular response against infection and has been reported to increase REE differently among patients with uncomplicated sepsis, sepsis syndrome, and septic shock (mean REE +55 ± 14%, +24 ± 12%, and +2 ± 24%) [26]. A higher REE in severe sepsis adult patients has also been associated with higher mortality [27], however further studies should investigate the effect of specific individual factors on metabolic evolution during sepsis and patient outcome.

#### Critical Illness

Critical illness is frequently associated with a hypermetabolic state related to the activation of catabolic hormones and resulting in elevated REE compared to healthy subjects. However, iatrogenic factors such as beta-blockers, analgesics, and sedatives may attenuate the response and even induce a hypometabolic state. Prolonged bed rest, atrophy of the metabolically active lean body mass, and mechanical ventilation have also been reported to decrease REE [1]. In patients with multiple organ failure, the loss of lean body mass is very rapid and resulted in 22% loss in 10 days [28] and no predictive equation showed good agreement compared to REE measured by IC [29].

Trauma patients showed hypermetabolism even when heavily sedated or medically paralyzed, however those with head injury on neuromuscular blockade or in a barbiturate coma showed reduced REE compared to similar patients without these agents [30]. Brain trauma was found to increase REE with high variability among different studies, ranging from 87% to 200% compared to predicted during the first 30 days post-injury; surprisingly, in patients admitted with a brain death diagnosis, the value ranged from 75% to 200% compared to predicted during the first 7 days [31]. 

Similarly, post-burn hypermetabolism was shown to increase REE as high as 100% above normal as consequence of a strong catabolic response mediated by endogenous catecholamines and inflammatory cytokines. Nevertheless, this hypermetabolic state is a direct effect of burn trauma, its persistence leads to severe septic complications, multiple organ failure, and higher mortality [16]. Nutritional follow up in burn patients shows a highly dynamic and variable REE up to 160 days after injury (Figure 3) [32].

The accurate determination of energy needs and the prevention of energy imbalance are essential in critically ill patients to avoid the harmful consequences of inadequate feeding. Underfeeding has been shown to increase hospital length of stay, infections, organ failure, to prolong mechanical ventilation, and to increase mortality, while overfeeding has been associated with hyperglycemia, hypertriglyceridemia, hepatic steatosis, azotemia, hypercapnia, and increased mortality [33]. In long staying patients with dynamic clinical conditions, IC should be repeated to monitor their nutritional requirements and avoid energy imbalance [34,35]. Optimal energy delivery targeting REE measured by IC seems to be significantly associated with reduced mortality, stressing the importance of this technique to assess caloric needs in ICU patients (Figure 4) [17].

The question which remains debated is the timing from which the measured REE should be prescribed as energy goal. Considering the importance of the early endogenous energy production, which is not suppressed by nutrition in the critically ill [36], feeding to measured value may result in overfeeding (Figure 5) [1]. Moreover, during the early phase of disease, catecholamines and the multiple treatments needed in these patients generate high instability and directly impact on measured REE. Tappy et al. showed that, in young starved trauma patients, the endogenous glucose production (EGP) was about 310 g/day (Figure 6), while the measured REE in these patients was 1830 kcal: feeding to the mean measured REE would clearly result in early overfeeding [37]. The figure also shows that endogenous glucose production continues in the sickest patients for many days even during feeding [8] (Figure 6). Therefore, the crude REE value provided by IC requires a careful interpretation to adequately prescribe exogenous energy supply. 

There is also a debate regarding the optimal timing to cover the energy value determined by IC and the capacity of the critically ill intestine to accommodate and absorb the delivered feeding. Gastrointestinal intolerance is frequent in ICU patients and may preclude the achievement of the predefined calorie target due to the incapability of absorbing supplied nutrition [38]. The best feeding route for each patient has to be assessed in order to limit the risk of stressing the intestine and cumulative caloric deficit [39].

### 3.2. Chronic Diseases

Energy requirements may be even more difficult to predict in patients with chronic conditions due to the large individual variations in REE. Both hyper- and hypo-metabolism have been shown in chronic pathologies due to alterations in metabolism, lean body mass, organ function, and presence of inflammation. The most common pathologies with important REE alterations are described below and summarized in Table 2. 

Both hyper- and hypo-metabolism, with respectively increased or decreased REE, have been observed in cancer patients. Several factors such as type, location, and size of tumor and presence of liver metastasis may contribute to this variability [16]. Similarly some studies showed increased REE in patients with chronic kidney disease, while others reported equal or even lower REE than those of matched healthy controls [40,41]. Renal replacement therapy such as hemodialysis or peritoneal dialysis also seem to affect REE, but results are contrasted [42,43]. Neurological diseases, such as Alzheimer’s, Parkinson’s, Huntington’s, and amyotrophic lateral sclerosis significantly affect patients’ REE as consequence of motor, endocrine, and metabolic abnormalities [44]. Diabetes is known to alter macronutrient metabolism, and increase the sympathetic activity leading to 5–10% augmentation of REE. On the contrary, the administration of anti-diabetic treatments is associated with a REE decrease [45]. Patients with chronic obstructive pulmonary disease feature an increased REE associated together with the progression of the disease severity, but a decreased TEE mainly due to a reduction of physical activity [46]. Both obesity and anorexia nervosa have been associated with altered REE mainly due to the different pattern of body composition compared to normal weight subjects. Obese patients showed significantly higher REE compared to non-obese patients; however, in most cases this difference disappeared after adjusting for fat-free mass [47]. Underweight and anorectic patients showed hypometabolic status with lower REE compared to predicted as a consequence of adaptation to starvation, loss of fat, and fat-free mass [48]. Conducting IC together with body composition measurement is useful to further optimize the nutrition prescription in these patients.

## 4. Benefits

The major benefit of performing IC in clinical practice is the prevention of both under- and over-feeding among patients with different conditions, thanks to the precise assessment and control of their energy needs. Inappropriate feeding has been associated with increased risk of infectious and non-infectious complications, length of hospitalization, more frequent readmission, and mortality, especially in ICU patients. Patients at risk of malnutrition were also found to be more likely to go to rehabilitation or nursing facilities rather than home after discharge [49]. In addition to negative outcomes in hospitalized patients, malnutrition also represents an economic burden for hospitals and the society [50]. In the ICU setting, supplemental parenteral nutrition (SPN) guided by IC has been shown to allow personalized and optimal nutrition in patients intolerant to full enteral nutrition. A reduction of nosocomial infections in the patients on SPN versus those on exclusive and insufficient enteral nutrition has been reported [5]. In depth analysis has further shown that the provision of SPN compared to exclusive enteral nutrition turned out to be a cost saving strategy [51,52]. Up to now, only four studies based their nutritional intervention on measured REE by IC [53]. A meta-analysis of available studies nevertheless shows that interventions guided by IC versus equations have significantly better clinical outcomes [34]. Promoting a systematic use of IC to individually optimize nutrition support among in- and out-patients, should result in meaningful clinical benefits and cost advantages. A large-scale study providing the benefits related to the use of IC to guide the prescription of nutrition support is needed.

## 5. Limitations

IC is the gold standard for measuring REE and for guiding nutrition support; however, several factors may limit the accuracy and/or the feasibility of the measurement (Table 3). During mechanical ventilation, air leakages in the respiratory circuit (particularly elevated in case of tracheostomy), high positive end expiratory pressure (PEEP > 10), fraction of inspired oxygen (FiO_2_) > 80%, and presence of other gases than O_2_, CO_2_, and N_2_ lead to unreliable results [1]. Thoracic drains are also a frequent cause of leakage in cardiothoracic units. Factors causing instability—such as agitation, fever, sedatives, and vasoactive adjustments during IC—limit the agreement between measured and real REE [54]. Likewise, organ support therapies such as continuous renal replacement therapy (CRRT) affect the respiratory pattern due to CO_2_ removal, therefore measured REE does not reflect patient’s real needs [54]. Similarly, to calculate real REE on extracorporeal membrane oxygenation (ECMO) patients requires gas exchange analysis by IC at both ventilator and ECMO sides [55,56].

Moreover, not all subjects are good candidates to IC (i.e., spontaneous breathing patients who are claustrophobic, nauseous, vomiting, or do not tolerate a face mask or being under a canopy cannot undergo IC measurement). Patients who require supplemental oxygen or undergoing noninvasive ventilation are also difficult to test due to limitations of software and techniques. Other restraints for performing IC in clinical routine are related to the equipment and maintenance cost, poor health insurance reimbursement, lack of trained manpower, difficulties interpreting the results, and lack of time to carry out the measurements [13].

## 6. Current Developments

Many indirect calorimeters are commercially available, but most of them are bulky, expensive and poorly accurate compared to a reference device [57,58,59,60]. Some indirect calorimeters use the ‘breath-by-breath’ technology and generate rapid readings by measuring short intervals of gas samples, but this method is error-prone due to the response time of gas analyzers and software. Others are equipped with a mixing chamber and offer more stable measurements, because the gas is physically ‘averaged’ before being analyzed. However, the presence of the mixing chamber (3–5 L) results in a large dimension device, which requires prolonged measurements (>20 min) to allow gas concentrations stability. This limits the possibility of doing short-time reliable measurements [57,58,59,60].

In recent years, hand-held calorimeters such as the MedGem™ and BodyGem™ (Microlife, Dunedin, FL, USA) have also been developed. While traditional indirect calorimeters measure VO_2_ and VCO_2_, the hand-held devices measure only VO_2_, while RQ is assumed to be 0.85. Advantages of these devices are portability, low degree of technical expertise required, and low cost. However, they cannot be used in ventilated patients, do not measure VCO_2_, and are not validated in hospitalized patients [61].

Indeed, several comparison studies have been conducted both in adults and children comparing standard metabolic carts to determine whether the MedGem™ device is accurate and reliable. The hand-held calorimeter showed discrepant results compared to Deltatrac^®^ in anorexia nervosa patients and healthy controls [62], overestimating REE by 8–11% [63] and agreeing only about 45% of the time in ambulatory adults [64]. MedGem^®^ showed agreement only 21% of the time with Vmax measures in outpatients with cirrhosis [65] and tended to overestimate REE in obese individuals [66]. Similarly, it was found inaccurate in children and adolescents with and without obesity, overestimating REE by 7–10% [67,68] compared to other indirect calorimeters such as Parvomedics^®^ and Deltatrac^®^, and underestimating by 8% [69] compared to Vmax^®^.

An international multicentric study (ICALIC), supported by two academic societies (ESPEN and ESICM), has recently led to the evaluation of a new generation calorimeter (Q-NRG^®^) developed to overcome the issue of inaccuracy, high cost, and lack of time [1]. It is equipped with a micro mixing chamber (2 mL) and features the technical characteristics required for measuring REE in both mechanically ventilated and spontaneous breathing subjects. In vitro evaluation showed gas analyzers’ accuracy within 2% difference against a mass spectrometer (MS) for the measurements of predefined gas mixtures and within 5% in a mechanically ventilated setting at oxygen enrichment up to 70% for measurements of simulated VO_2_ and VCO_2_ [70]. Q-NRG^®^ also showed very good accuracy and intra- and inter-unit precision in canopy mode both in vitro and in vivo compared to MS in healthy volunteers [71]. Studies to evaluate the ergonomics of the Q-NRG^®^ in the ICU and hospitalized patients are ongoing within the frame of ICALIC study: they compare the currently used calorimeters. In addition to accuracy and precision, the innovation and performances of this new calorimeter are reactivity, portability, battery-powered, affordability, monthly gas calibration, and warm-up free operations. Current innovations also led to the development of original devices to facilitate the personal monitoring of REE. An example is the Breezing^®^ device, a pocket-sized indirect calorimeter that measures oxygen consumption and carbon dioxide production rate in breath with a colorimetric technology. This device showed good agreement with the results from the Douglas Bag method for VO_2_, VCO_2_, EE, and RQ in healthy subjects but no tests were performed in clinical setting [72].

## 7. Alternative Methods to IC

So far, predictive equations remain the most common REE estimation method. They allow a rapid calculation of REE using anthropometric data (height, weight, sex, etc.) and have been validated among different group of hospitalized patients, but none is optimal (Table 4). Most of these equations have been developed in healthy subjects, resulting in large errors in case of critical illness and chronic diseases, despite the use of the correction factors [73]. Other methods for assessing REE have been explored and compared to IC in order to find a valid alternative. 

Fick’s principle uses cardiac output data, hemoglobin concentration, and arterial and mixed venous oxygen concentrations obtained from a pulmonary artery catheter to calculate REE. Comparison with IC in critically ill patients showed poor correlation with high variability between absolute values and variations of VO_2_ measured, unacceptable for clinical use [74,75]. The error found may be mostly related to five factors: the accuracy of the blood gas analyzer used to calculate arterial and venous oxygen content, the mixed of venous and bronchial arterial blood sample, the hemoglobin levels, the cardiac output variation over the respiratory cycle and the assumption of a fixed RQ to calculate REE [74,76]. 

The calculation of EE based on CO_2_ measurement collected from mechanical ventilators (EEVCO_2_) has also been proposed. This technique considers a fixed value of RQ to calculate the oxygen consumption (VO_2_) and REE through the Weir formula. Mainly due to the variability of RQ in critically ill patients, the accuracy of EEVCO_2_ compared to the Deltatrac^®^ metabolic monitor is poor [77,78,79,80].

The doubly labeled water method implies the oral administration of water with both hydrogen and oxygen atoms labeled with non-radioactive isotopes (^2^H/^1^H and ^18^O/^16^O) and uses their elimination rates in body liquids to calculate CO_2_ production. By assuming a fixed RQ this method allows calculating TEE in any subjects and any environment, and can been used in complementation to IC to calculate Physical Activity Level [81,82]. However, this technique is expensive and requires long time to get the results, limiting its use in daily clinical practice [3].

Motion sensors devices initially developed for fitness settings may be useful in clinical practice to monitor activity induced energy expenditure (AEE), in particular for patients undergoing physiotherapy during rehabilitation programs, as well as for the management of daily activity and the dietary programs in patients with obesity and/or diabetes. These devices are wearable on the arm, wrist, or waist; user-friendly; relatively low-cost; and non-invasive. Energy expenditure is derived from acceleration data and individual parameters (sex, age, weight, heart rate) using manufacturer’s confidential algorithms. However, the energy expenditure derived from these devices generally over- or underestimates the energy expenditure measured by IC by at least 10% [83]. Further studies are needed to validate their accuracy in clinical practice.

Measurements of body composition by bioelectrical impedance analysis (BIA) or dual energy X-ray absorptiometry (DXA) can be used to estimate REE. To this purpose predictive formula including FFM and FM values have been developed. This approach has been shown quite inaccurate in clinical populations compared to IC [73,84,85] and cannot be adopted in critically ill patients due to their abnormalities in hydration state and serum electrolyte concentrations that cause errors in the BIA-derived estimates of FFM and FM [86]. 

The agreement of predictive equations compared to IC results in critically ill patients was shown not to exceed 55%, especially in overweight or obese sarcopenic patients, and after prolonged physical immobilization [33,35,94]. Poor agreement is mainly due to the influence of body temperature, nutrition support, presence of sepsis, level of sedation, and therapies on REE [13]. The magnitude and the duration of the variations depend mostly on the severity of the disease with a considerable individual variability that cannot be assessed by equations [95]. Moreover, most of the equations base their calculations on anthropometric parameters, such as the body weight, which is often unknown and largely influenced by fluid retention or dehydration, leading to unreliable results [96].

Predictive equations tend to over (↑) or under (↓)—estimate REE in patients with chronic diseases such as:
chronic obstructive pulmonary disease (↓) [97]chronic kidney disease (↑) [98]amyotrophic lateral sclerosis (↓) [85]fibro-calculous pancreatic diabetes and diabetes type 2 (↓) [99]cancer (↑) [100]cirrhosis (↓) [101]

Similarly, predictive equations are inaccurate in patients with extreme BMI (BMI < 16 kg/m^2^ and BMI > 40 kg/m^2^) [102]. Errors are mainly due to an excessive or deficient fat mass, which is less metabolically active than the lean body mass, and to the body weight considered for the calculation (current, ideal, adjusted, or estimated) [73]. Therefore, it seems clear that standard factors to estimate the energy needs of individual patients are inappropriate and should be discouraged to guide nutrition support. Pragmatically, the ESPEN guideline suggests to use the simplest equation in critically ill patients—i.e., 20 kcal/kg/day during the first days—and in absence of IC to increase to 25 kcal/kg/day after 7 days [34]. 

## 8. Conclusions

The optimization of nutrition therapy is crucial for global patient care. In order to prevent under- and over-feeding and their related complications, it is important to accurately assess REE in individual patients and to ensure adapted nutrition support. IC is considered as the gold standard to this purpose and ideally any patients in whom energy needs are uncertain should be measured. Recent developments should facilitate the widespread use of IC in medical routine and promote better clinical outcomes. Considering the ongoing debate, the widespread use of IC might finally enable the design of prospective studies which will be able to determine the optimal dose of energy to deliver during the different stages of disease, i.e., the ratio of the energy delivered to measured REE and timing of feeding.

## Figures and Tables

**Figure 1 jcm-08-01387-f001:**
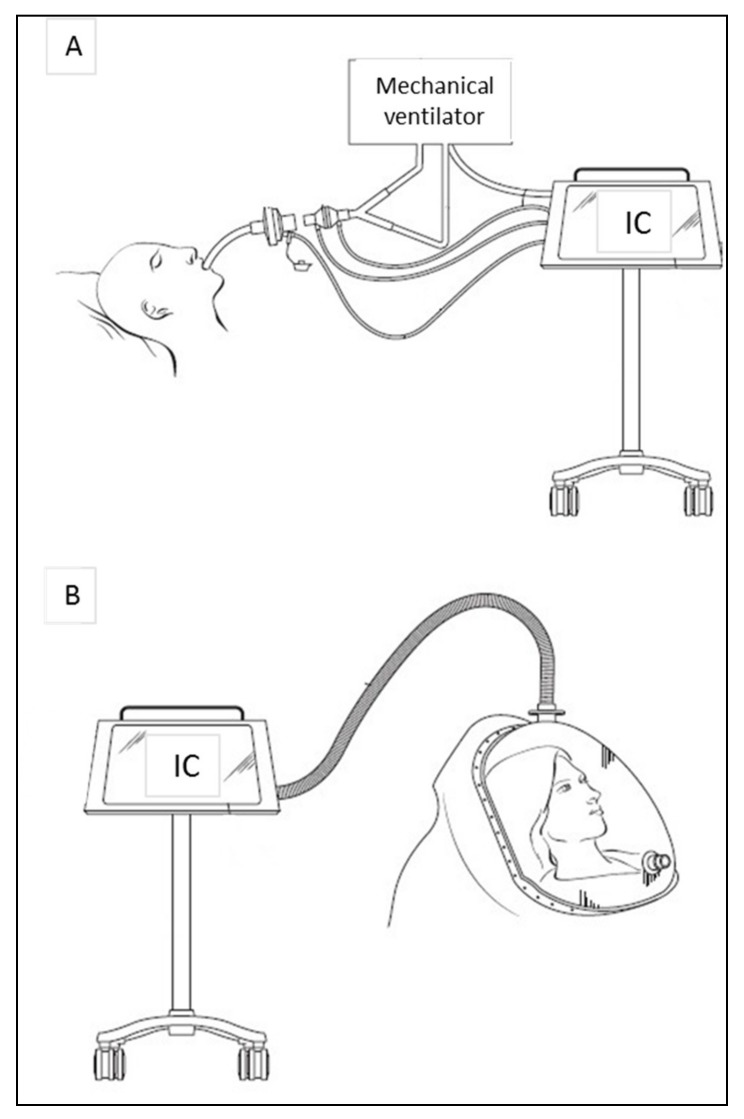
Indirect calorimetry on mechanically ventilated patient (**A**) and on spontaneous breathing patient in canopy mode (**B**). In mechanical ventilation the gas sampling is acquired by the circuit connecting the endotracheal tube to the ventilator and measured by ‘breath-by-breath’ or mixing chamber analyses. In spontaneous breathing mode, the subject is placed under a clear canopy with a plastic drape to avoid air leakage. Breath exchanges are collected by the calorimeter for gas analysis and enable calculation of REE using Weir’s equation (REE (kcal/day) = [(VO_2_ × 3.941) + (VCO_2_ × 1.11)] × 1440).

**Figure 2 jcm-08-01387-f002:**
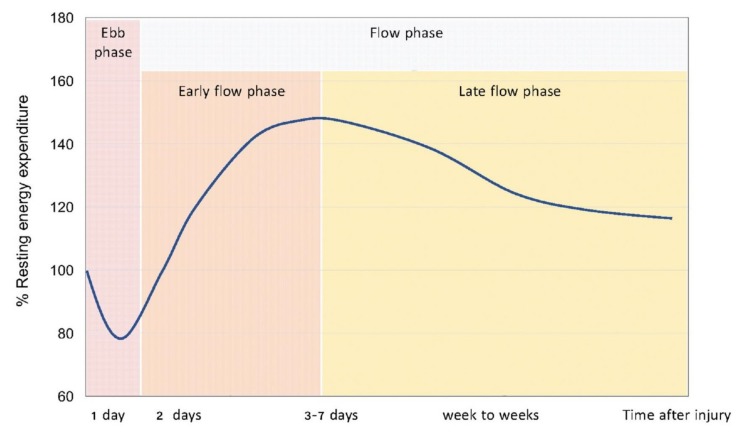
Metabolic response to injury proposed by Cuthbertson et al. A short ebb phase characterized by hypometabolism occurs immediately after the injury and is characterized by a decrease in metabolic rate, oxygen consumption, body temperature, and enzymatic activity. The ebb phase is followed by a longer hypermetabolic flow phase marked by an increased catabolism, with a high oxygen consumption and an elevated REE rate. Reused with permission from [16].

**Figure 3 jcm-08-01387-f003:**
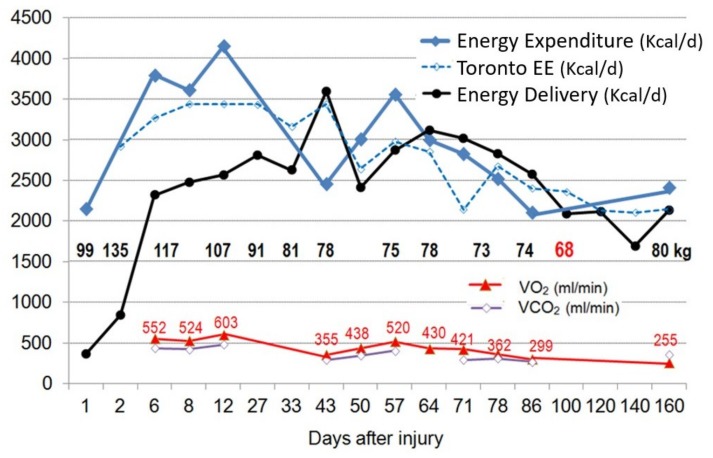
Evolution of measured REE by IC (blue), Toronto predictive equation (dashed blue), delivered energy (black), VO_2_ (red △), and VCO_2_ (purple ◆) in a young man weighing 99 kg upon admission with major burns covering 85% body surface over 160 days. The REE variations were important over time particularly during the early phase (weight gain due to fluid resuscitation was 36 kg by day 3), and paralleled the loss of body weight, i.e., of lean body mass (−31 kg after 3 months, with slow recovery). The REE value on day 1 corresponds to the Harris & Benedict prediction of basal EE. The figure also shows the reasonable precision of the Toronto equation, and how difficult it is to feed to measured IC value during the first 14 days. Adapted from [32].

**Figure 4 jcm-08-01387-f004:**
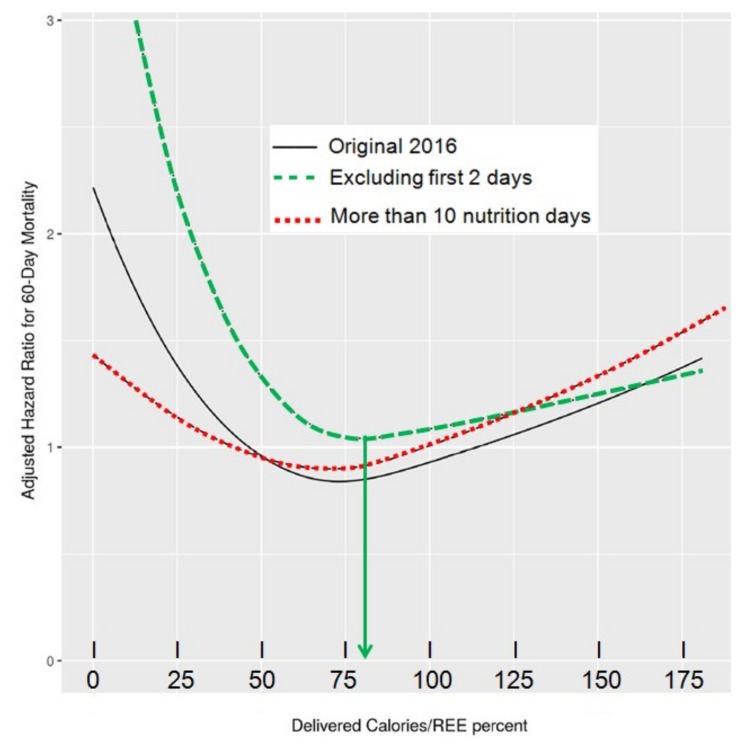
Association of delivered calories/resting energy expenditure (REE) percent by indirect calorimetry (IC) with 60-day mortality in different models: the authors recalculated their original 2016 data to integrate the fact that energy delivery increased progressively during the initial 2–3 days, reducing the mean value in stays <5 days. The lowest ICU mortality was observed when percent of delivered calories by REE obtained by IC was 80% (excluding first two feeding days) and 75% (with >10 evaluable nutrition days) (*p* < 0.05). On the contrary, increments of the ratio above that point—specifically >110%—were associated with increasing mortality (*p* < 0.05). Reproduced with permission (http://creativecommons.org/licenses/by/4.0/) [17].

**Figure 5 jcm-08-01387-f005:**
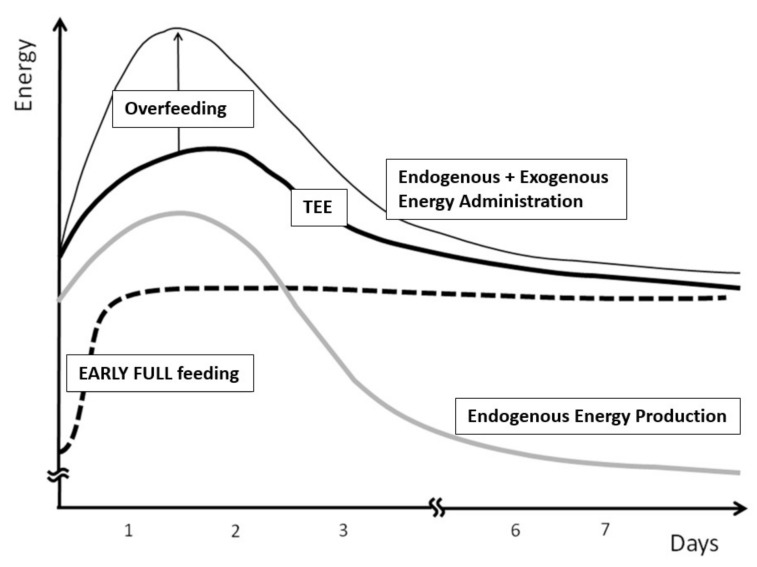
Conceptual representation of the relative overfeeding commonly resulting from early full feeding during the first days of critical illness. During this phase, the endogenous glucose production (EGP) is increased, covering up to two-thirds of total energy expenditure (TEE—solid black bold line). Full feeding in this phase will results in overfeeding, as the EGP is not attenuated by energy administration (different form healthy): exogenous feeding adds to the EGP resulting in an excessive energy availability, superior to TEE. (Solid black bold line: TEE; grey bold line: adapted endogenous energy production; dotted bold line: early energy administration; thin line: combined endogenous and exogenous energy administration). Reproduced with permission from [1].

**Figure 6 jcm-08-01387-f006:**
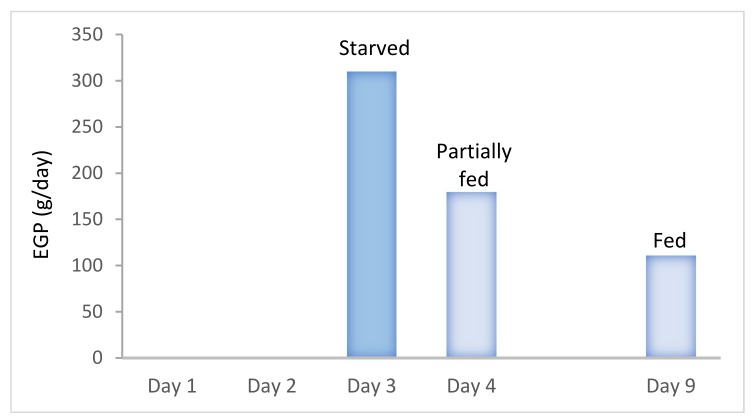
Evolution of the endogenous glucose production (EGP) over time in critically ill patients. EGP was shown to be 310 g/day in 40 years old starved trauma patients at day 3 of ICU admission (blue) [37] and then decreased to 180 and 111 g/day in older fed patients at day 4 and 9 respectively (light blue) [8]. EGP has to be considered as source of energy in order to avoid overfeeding by extrinsic energy during the first days of ICU stay. Data combined from [8,37].

**Table 1 jcm-08-01387-t001:** Factors influencing resting energy expenditure. Adapted from [1,2,9].

Effects on REE	Factors	
**↑**	Burns Hyperventilation Hyperthermia Hyperthyroidism, pheochromocytoma Inflammation (interleukins, interferons, tumor necrosis factors etc.) Metabolic acidosis	Morbid obesity Overfeeding Physical agitation Sepsis Stress (epinephrine, cortisol, glucagon etc.)
**↓**	Coma/deep sleep General anesthesia Heavy sedation Hypothermia Hypothyroidism Hypoventilation	Gluconeogenesis Metabolic alkalosis Paralysis Sarcopenia, cachexia Starvation/underfeeding/ketosis

**Table 2 jcm-08-01387-t002:** Common chronic pathologies with effects on resting energy expenditure

Condition	Effect on REE	
Anorexia nervosa	↓	Low energy intake and reduced lean body mass
Cancer	↑ ↓	Cancer growth and inflammation Progressive reduction of lean body mass
Chronic kidney diseases	↑ ↓	Metabolic acidosis and inflammation Acute and chronic renal failure
Chronic obstructive pulmonary disease	↑	Increased respiratory efforts
Diabetes	↑	Increased metabolism
Obesity	↑ ↓	Increased lean body mass Sarcopenia
Neuromuscular degenerative diseases	↑ ↓	Inflammation and endocrine disorders Dysfunction of muscle tissue

**Table 3 jcm-08-01387-t003:** Factors limiting the reliability and feasibility of IC measurement.

Factors Limiting IC Measurement
Agitation, fever, sedatives, and vasoactive adjustments during measurement
Air leakages in respiratory circuit
Dialysis or continuous renal replacement therapy
ECMO
Mechanical ventilation with PEEP > 10 Mechanical ventilation with FiO_2_ > 80% Noninvasive ventilation Other gases than O_2_, CO_2_, and N_2_: helium
Supplemental oxygen in spontaneous breathing patients

**Table 4 jcm-08-01387-t004:** Some predictive equations commonly used in clinical practice and tested among different hospitalized patients against IC.

Equations	Parameters Used for Calculation	Accuracy Rate *
**General Hospitalized Population**		
25 kcal/kg	25 × WT	43% [10] 23% [87]
Harris & Benedict (1919)	M: 13.75 × WT + 5.00 × HT − 6.75 × age + 66.47 F: 9.56 × WT + 1.85 × HT − 0.67 × age + 655.09	43% [10] 38% [87]
Ireton-Jones (1992)	1925 − 10 × age + 5 × WT + (281 if male) + (292 if trauma) + (851 if burn)	28% [10]
Mifflin-St Jeor (1990)	M: 10 × WT + 6.25 × HT − 5 × age + 5 F: 10 × WT + 6.25 × HT − 5 × age − 161	35% [10] 32% [87]
Schofield (1985)	8.4 × WT + 4.7 × HT + 200	42% [87]
**Anorexic Patients (BMI < 16)**		
Bernstein et al. (1983)	M: 11.02 × WT + 10.23 × HT − 5.8 × age − 1032 F: 7.48 × WT − 0.42 × HT − 3 × age + 844	40% [73]
Harris & Benedict (1919)	M: 13.75 × WT + 5.00 × HT − 6.75 × age + 66.47 F: 9.56 × WT + 1.85 × HT − 0.67 × age + 655.09	39% [73]
Huang et al. (2004)	10.16 × WT + 3.93 × HT − 1.44 × age + 273.82 × sex + 60.65	43% [73]
Lazzer et al. (2007)	M: 0.05 × WT + 4.65 × HT − 0.02 × age − 3.60 F: 0.04 × WT + 3.62 × HT − 2.68	39% [73]
Mifflin-St Jeor (1990)	M: 10 × WT + 6.25 × HT − 5 × age + 5 F: 10 × WT + 6.25 × HT − 5 × age − 161	40% [73]
Müller et al. (2004)	0.05 × WT + 1.01 × sex + 0.015 × age + 3.21	37% [73]
Owen (1987)	M: WT × 10.2 + 879 F: WT × 7.18 + 795	41% [73]
**Obese Patients (BMI > 30)**		
Bernstein et al. (1983)	M: 11.02 × WT + 10.23 × HT − 5.8 × age − 1032 F: 7.48 × WT − 0.42 × HT − 3 × age + 844	16% [88] 21% [89]
Harris & Benedict (1919)	M: 13.75 × WT + 5.00 × HT − 6.75 × age + 66.47 F: 9.56 × WT + 1.85 × HT − 0.67 × age + 655.09	64% [88]
Huang et al. (2004)	10.16 × WT + 3.93 × HT − 1.44 × age + 273.82 × sex + 60.65	66% [88] 53% [89] 54% [90]
Lazzer et al. (2007)	M: 0.05 × WT + 4.65 × HT − 0.02 × age − 3.60 F: 0.04 × WT + 3.62 × HT − 2.68	58% [88] 46% [90]
Mifflin-St Jeor (1990)	M: 10 × WT + 6.25 × HT − 5 × age + 5 F: 10 × WT + 6.25 × HT − 5 × age − 161	52% [89] 56% [90]
Müller et al. (2004)	0.05 × WT + 1.10 × sex + 0.016 × age + 2.92	60% [88] 58% [89] 47% [90]
Owen (1987)	M: WT × 10.2 + 879 F: WT × 7.18 + 795	38% [73] 40% [89]
**Critically Ill Patients**		
25 Kcal/Kg	25 × WT	12% [91]
Harris-Benedict (1919)	M: 13.75 × WT + 5.00 × HT − 6.75 × age + 66.47 F: 9.56 × WT + 1.85 × HT − 0.67 × age + 655.09	31% [91] 32% [92]
Ireton-Jones (1997)	1925 − 10 × age + 5 × WT + (281 if M) + (292 if trauma) + (851 if burn)	37% [93]
Mifflin-St Jeor (1990)	M: 10 × WT + 6.25 × HT − 5 × age + 5 F: 10 × WT + 6.25 × HT − 5 × age − 161	18% [91] 35% [10]
Owen (1987)	M: WT × 10.2 + 879 F: WT × 7.18 + 795	12% [91]
Penn State (2003)	0.85 × HB + 175 × Tmax + 33 × Ve − 6433	43% [10]
Swinamer (1990)	945 × BSA − 6.4 × age + 108 T + 24.2 × RR + 81.7 × VT − 4349	55% [93] 45% [10]

* % of patients where the predicted value (by equation) is within 10% of measured value (by IC). BSA, body surface area (m^2^); HB, Harris–Benedict value; HT, height (cm); RR, respiratory rate (breath/min); sex: males (M) = 1, female (F); T, temperature (°C); Tmax, maximum temperature (°C) in previous 24 h; TV, tidal volume (L); Ve, expired minute ventilation at the time of collection (L/min); WT, weight (kg).

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
