# Peer review of "Indirect Calorimetry in Clinical Practice"

_jcm, 2019, doi:10.3390/jcm8091387_

Round 1
Reviewer 1 Report
General comments
The paper by Delsoglio et al., is a review aimed at summarizing the concepts and application of indirect calorimetry in clinical practice.
The topic is extremely interesting because a precise evaluation of energy needs and substrate utilization pattern could be very important in order to avoid the well known complications of both over- and underfeeding especially when artificial nutrition is needed.
The review is clear, well written, and the concepts are well presented and discussed, also for the non expert reader. The advances in the field as well as the future perspectives of the application of IC and new simplified devices are well presented.
The review will make available for the reader a useful tool to inform clinical practice and decision making.
Minor remarks
In order to further improve the completeness of the paper, I think that Acute Kidney Injury, a condition quite frequently observed among ICU patients, should be more in depth discussed. The Authors correctly cited (page 4) an old and fundamental paper published by Schneweiss in 1990 in Am J Clin Nutr, but there are now some more recent papers that could be cited and they data discussed:
Hellerman M, et al. Carbohydrate and Lipid Prescription, Administration, and Oxidation in Critically Ill Patients With Acute Kidney Injury: A Post Hoc Analysis. J Ren Nutr. 2019;29(4):289-294 30630662. Góes CR, et al., Evaluation of Factors Associated with Hypermetabolism and Hypometabolism in Critically Ill AKI Patients. Nutrients 2018;10(4). pii: E505 Sabatino A, et al., Energy and Protein in Critically Ill Patients with AKI: A Prospective, Multicenter Observational Study Using Indirect Calorimetry and Protein Catabolic Rate. Nutrients 2017; 26;9(8). pii: E802. doi: Góes CR, et al., Influence of different dialysis modalities in the measurement of resting energy expenditure in patients with acute kidney injury in ICU. Clin Nutr 2017; 36(4):1170-1174
Author Response
General comments
The paper by Delsoglio et al., is a review aimed at summarizing the concepts and application of indirect calorimetry in clinical practice.
The topic is extremely interesting because a precise evaluation of energy needs and substrate utilization pattern could be very important in order to avoid the well known complications of both over- and underfeeding especially when artificial nutrition is needed.
The review is clear, well written, and the concepts are well presented and discussed, also for the non expert reader. The advances in the field as well as the future perspectives of the application of IC and new simplified devices are well presented.
The review will make available for the reader a useful tool to inform clinical practice and decision making.
We would like to thank the reviewer for the nice and encouraging comment.
Minor remarks
In order to further improve the completeness of the paper, I think that Acute Kidney Injury, a condition quite frequently observed among ICU patients, should be more in depth discussed. The Authors correctly cited (page 4) an old and fundamental paper published by Schneweiss in 1990 in Am J Clin Nutr, but there are now some more recent papers that could be cited and they data discussed:
Hellerman M, et al. Carbohydrate and Lipid Prescription, Administration, and Oxidation in Critically Ill Patients With Acute Kidney Injury: A Post Hoc Analysis. J Ren Nutr. 2019;29(4):289-294 30630662. Góes CR, et al., Evaluation of Factors Associated with Hypermetabolism and Hypometabolism in Critically Ill AKI Patients. Nutrients 2018;10(4). pii: E505 Sabatino A, et al., Energy and Protein in Critically Ill Patients with AKI: A Prospective, Multicenter Observational Study Using Indirect Calorimetry and Protein Catabolic Rate. Nutrients 2017; 26;9(8). pii: E802. doi: Góes CR, et al., Influence of different dialysis modalities in the measurement of resting energy expenditure in patients with acute kidney injury in ICU. Clin Nutr 2017; 36(4):1170-1174
Thank you very much for this comment. Indeed there are more recent data, we added this information in the text. L109-114
Reviewer 2 Report
Overall the review is very comprehensive and well-structured. It will add to the current evidence in this area. Well done.
Author Response
Comments and Suggestions for Authors
Overall the review is very comprehensive and well-structured. It will add to the current evidence in this area. Well done.
We would like to thank the reviewer for the kind and encouraging comment.
Reviewer 3 Report
This review aims to summarize the latest innovations of Indirect Calorimetry, the clinical indications for IC, and the benefits and the limitations of IC in clinical settings and conditions. This review is very thorough on conditions that effect REE, therefore, making a strong case for IC.The review is not as thorough examining the different methods for measuring IC and while it points out problems with IC, it doesn't indicate any best practices or how to do IC in clinical conditions to get the best and most accurate results.
While I don't have specific recommendations, I'd recommend trying to streamline, focus, and shorten most of the manuscript. I'd also recommend expanding the current developments and adding a section on best practices.
Specific primarily grammatical changes include:
Line 47 change at summarizing to to summarize Lines 73 and 83 Weir’s Line 74 More of most from Line 107 change as effect to as the effect Line 114-115 change with a higher mortality to either with higher mortality or with a higher mortality rate Line 129-130 of 75% predicted in patients admitted with a brain death diagnosis This statement is not clear. Is it 75% of normal or is 75% higher than normal. Reword to make it clear what you mean. Line 167-168 The question which remains debated is the timing of making the application of the measured REE the as energy goal. Line 200 40 years old Line 212 type, location, and size of tumor Line 221 with a REE decrease Line 222 REE associated together with Line 223 Either Both obesity or anorexia Line 236-237 more frequent readmission and increased mortality, especially in ICU patients. Line 243-244 Reword this sentence. I’m not sure what is is supposed to say In depth analysis has further shown that SPN, in spite of its complexity , to be superior to exclusive enteral nutrition andturned out to be a cost saving strategy Line 269 lack of trained manpower, difficulties to interpreting the results, and lack of time to carry out the measurements [55]. Line 276 poorly inaccurate Line 281 large dimension device and that requires prolonged measurements Line 324 The Fick’s principle Line 343 Motion sensors devices Lines 352-355 Measurements of body composition by bioelectrical impedance analysis (BIA) or dual energy X-ray absorptiometry (DXA) can be used to derive estimate To this purpose predictive formula including FFM and FM values have been developed. This approach has been shown quite inaccurate in clinical populations compared to IC Line 365-366 Accuracy of predictive equations in critically ill patients was shown not to exceed 50% compared to IC results, What does this mean—it’s within 10% accuracy is less than 50% of the cases. Line 383 Errors are mainly due Line 387 Pragmatically the ESPEN guideline suggests to use the simplest equation in critically ill Line 391 The optimization of nutrition therapy is crucial for the global patient’s care. Odd choice of words Lin 396-397 Considering the ongoing debate Widespread use of IC could then lead to prospective studies which will then be able to determine test the optimal dose, i.e. the ratio of the energy delivered to measured REE, and timing of feeding.
Author Response
Comments and Suggestions for Authors
This review aims to summarize the latest innovations of Indirect Calorimetry, the clinical indications for IC, and the benefits and the limitations of IC in clinical settings and conditions. This review is very thorough on conditions that effect REE, therefore, making a strong case for IC. The review is not as thorough examining the different methods for measuring IC and while it points out problems with IC, it doesn't indicate any best practices or how to do IC in clinical conditions to get the best and most accurate results.
While I don't have specific recommendations, I'd recommend trying to streamline, focus, and shorten most of the manuscript. I'd also recommend expanding the current developments and adding a section on best practices.
Thank you for your recommendations and suggestions: all of the below comments have been addressed. But at this stage reorganizing completely the paper as suggested would require many days which is beyond the time we have for revision.
Specific primarily grammatical changes include
Line 47 change at summarizing to to summarize
Lines 73 and 83 Weir’s
Line 74 More of most from
Line 107 change as effect to as the effect
Line 114-115 change with a higher mortality to either with higher mortality or with a higher mortality rate
Line 129-130 of 75% predicted in patients admitted with a brain death diagnosis This statement is not clear. Is it 75% of normal or is 75% higher than normal. Reword to make it clear what you mean.
Line 167-168 The question which remains debated is the timing of making the application of the measured REE the as energy goal.
Line 200 40 years old
Line 212 type, location, and size of tumor
Line 221 with a REE decrease
Line 222 REE associated together with
Line 223 Either Both obesity or anorexia
Line 236-237 more frequent readmission and increased mortality, especially in ICU patients.
Line 243-244 Reword this sentence. I’m not sure what is is supposed to say In depth analysis has further shown that SPN, in spite of its complexity , to be superior to exclusive enteral nutrition and turned out to be a cost saving strategy
Line 269 lack of trained manpower, difficulties to interpreting the results, and lack of time to carry out the measurements [55].
Line 276 poorly inaccurate
Line 281 large dimension device and that requires prolonged measurements
Line 324 The Fick’s principle
Line 343 Motion sensors devices
Lines 352-355 Measurements of body composition by bioelectrical impedance analysis (BIA) or dual energy X-ray absorptiometry (DXA) can be used to derive estimate To this purpose predictive formula including FFM and FM values have been developed. This approach has been shown quite inaccurate in clinical populations compared to IC
Line 365-366 Accuracy of predictive equations in critically ill patients was shown not to exceed 50% compared to IC results, What does this mean—it’s within 10% accuracy is less than 50% of the cases.
Line 383 Errors are mainly due
Line 387 Pragmatically the ESPEN guideline suggests to use the simplest equation in critically ill
Line 391 The optimization of nutrition therapy is crucial for the global patient’s care. Odd choice of words
Lin 396-397 Considering the ongoing debate Widespread use of IC could then lead to prospective studies which will then be able to determine test the optimal dose, i.e. the ratio of the energy delivered to measured REE, and timing of feeding.
We would like to thank the reviewer for a very careful and high quality review which has helped us improve our paper. We modified the text accordingly to the suggestions.
This manuscript is a resubmission of an earlier submission. The following is a list of the peer review reports and author responses from that submission.
Round 1
Reviewer 1 Report
This review summarizes the latest innovations of IS as wel as the clinical indications, the benefits and the limitations.
Major comment
This review is an overview from most what we already know and concerning the latest innovations only the paragraph about current developments describes some new issues. The manuscript is quite similar to the manuscript of Oshima et al Clin Nutr 2017
Additional comments:
Introduction
Line 23: one can debate about the word precise
Line 29: 60-65% of REE of the total energy espenditure; this is in healthy adults and for example not in children and extreme obese
Line 42: reference 3 is not referring to trials
Indications
Line 52: it is stated that IC is indicated in acute critical illness associated with large and dynamic changes of metabolic stress level; this is really debatable
Line 55: reference 7 is from 1989, nothing new?
Line 58: the authors write: avoid complications of inappropriate nutrition: like what?
Line 59: please give references by all these factors from the most recent literature
Practicalities
Line 85: are there any studies who have repeated these measurements?
Line 97: the auhtors state that IC is a fundemental tool…..this is really the question and should be explained more in detail
Line 107: ref 14 is from 1993 and the REE differs between sepsis, sepsis syndrome and septic shock markedly. Any other explanation tha the hyperdynamic response? Has this never repeated afterwards? Is figure 2 still valid, why not make a figure with all studies done in critical illness on different days?
Line 129: I don’t see the additional value of this figure, it is also not a clear and illustrating figure
Line 134: this sentence is a repeat of sentences before
Line 143: ref 23 is a retrospective cohort study so the conclusion of this study has to judged with caution
Line 158: make a figure with the contribution of endogenous energy production compared to REE during the course of critical illness
Line 168: please add in table 2 the ranges of REE which are measured. In table 2 I don’t understand the reasons which are mentioned for increased or decreased REE. Which factors are related also to increased REE and which ones to decreased REE
Line 203: why is only the SPN study mentioned concerning delivering of PN. It is also the question if the use of IC has given these results of the trial.
Line 201: ref 21 is a guideline and nog a meta-analysis
Line 226: please provide the reader with a table of limitations in using IC. The most sickest patients can’t be measured and for example those on ECMO are extremely difficult to measure
When IC is not available
Line 268: table 3 has no additional value
Line 273-line 296: can be summarized to 1 sentence
What about VCO2 measurements on the ventilator?
Reviewer 2 Report
This well-written paper aims to provide a narrative review of the clinical indications, benefits, limitations and latest innovations of indirect calorimetry.
Overall, the topic is clearly presented, scientifically sound and up to date. Importantly, the content does not overlap that of another review published earlier this year nearly on the same topic (Rattanachaiwong S et al. Clinical Nutrition. 2019).
I simply suggest a few edits to the Figures to improve clarity:
i. Figure 3:
- Please define the green symbol in the caption.
- Units on the Y-axis are confusing. Also, kg is indicated both on the Y-axis and the graph. If possible, I suggest moving the units to the legend.
ii. Figure 4 is difficult to understand based on the caption. Title of X-axis is missing and its formatting needs to be adjusted in the final version.
Reviewer 3 Report
The authors present a very valuable review of energy metabolism in clinical practice, with the emphasis for the application and value of indirect calorimetry.
While this reviewer appreciates the review of different metabolic conditions on 'resting' metabolic rate, the limitations of indirect calorimetry are inadequately discussed.
The main concern is the translation of measured energy expenditure into energy intake recommendations. The authors dedicate a paragraph towards the increased endogeneous energy supply by the liver, which adds significant uncertainty to the energy intake requirement. However, the authors do not discuss the ability/limitations of gastrointestinal tract to absorb the energy supplied. For example, if energy expenditure is increased, can the GI-tract absorb adequate energy in conditions of severe illness/shock?
Second, the recent developments of new, portable indirect calorimetry measurements should be elaborated on. What are their accuracies (in clinical populations, per individual)? In that respect, the current focus of the manuscript seems to be the discussion of various conditions on REE rather than the methodology and its applicability in these populations.
Further, IC, specifically portable IC, should be compared to alternative methods to determine energy requirements, e.g. titrating body weight, stable isotopes, BIA.
Minor comments include
page 1, para 1: variability in REE across individuals range from 40/50 to 80%
protein oxidation cannot be assesses by IC without urine collection
age and sex have effects on REE
hyper/hypo-metabolism are defined by REE, and not as having an effect of REE!?